# Smoking Impairs Hematoma Formation and Dysregulates Angiogenesis as the First Steps of Fracture Healing

**DOI:** 10.3390/bioengineering9050186

**Published:** 2022-04-24

**Authors:** Helen Rinderknecht, Andreas K. Nussler, Konrad Steinestel, Tina Histing, Sabrina Ehnert

**Affiliations:** 1Siegfried-Weller Institute for Trauma Research, BG Trauma Center, University of Tuebingen, Schnarrenbergstrasse 95, 72070 Tuebingen, Germany; helen.rinderknecht@student.uni-tuebingen.de (H.R.); andreas.nuessler@gmail.com (A.K.N.); thisting@bgu-tuebingen.de (T.H.); 2Institute of Pathology and Molecular Pathology, Bundeswehrkrankenhaus Ulm, Oberer Eselsberg 40, 89081 Ulm, Germany; konradsteinestel@bundeswehr.org

**Keywords:** Trauma, bone, fracture repair, smoking, tissue engineering, in vitro, angiogenesis

## Abstract

Bone fracture healing is an overly complex process in which inflammation, osteogenesis, and angiogenesis are tightly coupled, and delayed fracture repair is a very common health risk. One of the major causes of delayed healing is the formation of insufficient vasculature. Precise regulation of blood vessels in bone and their interplay with especially osteogenic processes has become an emerging topic within the last years; nevertheless, regulation of angiogenesis in (early) diseased fracture repair is still widely unknown. Here, we aim to develop an in vitro model for the analysis of early fracture healing which also enables the analysis of angiogenesis as a main influencing factor. As smoking is one of the main risk factors for bone fractures and developing a delay in healing, we model smoking and non-smoking conditions in vitro to analyze diverging reactions. Human in vitro fracture hematomas mimicking smokers’ and non-smokers’ hematomas were produced and analyzed regarding cell viability, inflammation, osteogenic and chondrogenic differentiation, and angiogenic potential. We could show that smokers’ blood hematomas were viable and comparable to non-smokers. Smokers’ hematomas showed an increase in inflammation and a decrease in osteogenic and chondrogenic differentiation potential. When analyzing angiogenesis, we could show that the smokers’ hematomas secrete factors that drastically reduced HUVEC proliferation and tube formation. With an angiogenesis array and gene expression analysis, we could identify the main influencing factors: Anpgt1/2, Tie2, and VEGFR2/3. In conclusion, our model is suitable to mimic smoking conditions in vitro showing that smoking negatively impacts early vascularization of newly formed tissue.

## 1. Introduction

Delayed fracture healing is one of the main causes of prolonged hospital stays. It is a burden for the patient and additionally produces large costs for the health care system [1]. Between 10 and 20% of all long bone fractures show impaired healing or even result in fracture non-unions. Impaired healing is often associated with the patient’s aberrant inflammatory status, e.g., caused by smoking, type II diabetes, or medication. Additional known risk factors are age, gender, or excessive alcohol consumption [2,3]. Smoking is still one of the main risk factors, and even though negative health risks of smoking are widely known, it is still a widespread habit in the population [4]. Even in 2020, one in four Germans aged 15 and older consumed cigarettes on a daily basis (OCED average 16.5%) [5]. Smoking patients not only show higher rates of fracture non-unions and complications, but also an increased risk of fracture due to lower bone mass [6,7].

Intact vasculature is mandatory for physiological tissue homeostasis by transporting oxygen, nutrients, and metabolic waste products [8]. The fracturing of the bone leads to the rupture of the neighboring blood vessels and subsequently the formation of a fracture hematoma within the fracture gap. The rupture of the vessels leads to a sudden separation from the oxygen supply and subsequently a hypoxic environment. Proper fracture hematoma formation has been reported to be essential for successful bone healing [9]. Primarily, the hematoma leads to a stabilization of the fracture ends [7,8]. However, its major task is the orchestration of the inflammatory phase of healing, in which the ordered secretion of proinflammatory cytokines (amongst others, Interleukin (IL)-1β, IL-8, IL-6, Tumor Necrosis Factor-alpha (TNF-α), C-C motif chemokine ligand 2 (CCL2)) and growth factors (including Transforming growth factor-beta (TGF-β), Platelet-derived growth factor (PDGF), Fibroblast growth factor (FGF), Insulin-like growth factor 1 (IGF-I)) by various resident and invading cells leads to recruitment of first inflammatory and immune cells followed by mesenchymal stem or osteoprogenitor cells and endothelial (progenitor) cells to the site of fracture [10]. Revascularization of the newly formed callus is extremely important for the ongoing healing process, and insufficient vasculature of the fracture callus is a major cause for non-unions [11]. Hematomas have been shown to develop angiogenic as well as osteogenic potential within 3 to 4 days, which can partially be attributed to the specificity of the fracture microenvironment [12]. During the inflammatory phase, detailed information about angiogenesis in bone fracture repair is still limited. Several studies have shown the importance of growth factors of different families such as the VEGF, FGFs, Angiopoietins, or BMPs, whereas VEGF is by far the best-studied one and of eminent relevance [13,14]. The growth factor is essential during normal fracture repair by stimulating endothelial cells to form new vessels, and by assisting in the communication between the endothelium and the bone [15]. Within the fracture gap, the drop in oxygen initiates HIF-1α-guided VEGF production, mainly by invading macrophages and adjacent hypertrophic chondrocytes [16,17].

Treatment to avoid a delay in fracture repair usually combines mechanical stabilization and biological stimulation using cellular bone matrices, mainly focusing on promoting osteogenesis. More advanced matrices are also designed to deliver growth factors such as recombinant human BMP-2, PDGF-BB, or VEGF to the site of fracture. Nevertheless, these matrices are still not broadly used for therapy [18].

Within this study, we want to analyze the effect of smoking on early fracture repair in vitro, as it is one of the main risk factors for the development of fracture healing disorders. Cigarette smoke consumption not only interferes with the inflammatory status by, for instance, increasing oxidative stress and altered cytokine secretion, but also influences bone metabolism via toxic components of the smoke, oxygenation, altered hormone status, etc. [6,19]. The vascular system is also severely affected by smoke and smoking is the most important modifiable risk factor for the development of cardiovascular diseases [20]. Smoking increases oxidative stress in endothelial cells, platelet aggregation, and arteriosclerosis and reduces the amount of bioavailable nitric oxide, which is associated with a reduction in the flow-mediated dilation of the vessels [21].

For our analysis, we modified an in vitro fracture hematoma model [22,23] to display smoking and non-smoking early fracture repair with additional consideration of the vascular system. We investigated whether our model can be used to recreate previously published results on the inflammatory status and osteogenic/chondrogenic differentiation of MSCs in smokers as a more complex representation of early fracture repair. As a mandatory contributor to appropriate healing, we further identified if and how angiogenic stimuli and angiogenesis in smokers are altered. Therefore, smokers’ and non-smokers’ in vitro hematomas were produced and compared regarding viability, inflammation, osteogenic potential, and angiogenesis.

## 2. Materials and Methods

### 2.1. Culture of SCP-1 Cells

Human immortalized bone marrow-derived mesenchymal stem cell (hB-MSCs) line SCP-1 was obtained from Prof. Schieker. SCP-1 cells were immortalized by hTERT lentiviral gene transfer, which prevented senescence by maintaining all essential functions including the differentiation ability [24]. A transduced MSC cell line was used in these experiments to increase comparability and due to the high demand of cells. Cells were cultured in Minimal Essential Medium Alpha (MEM-α, HiMedia Laboratories, Mumbai, India) containing 5% Fetal calf serum (FCS, Thermo Scientific, Waltham, MA, USA). The medium was changed every 3 to 4 days. Cells were subcultured when reaching confluency. For experiments, cells were used in passages between 4 and 15.

### 2.2. Preparation of In Vitro Hematomas and Culture

Hematomas were produced following Pfeiffenberger et al. 2019 [22]. Therefore, a 1 × 10^6^ cells/mL cell suspension of SCP-1 cells in clotting medium (MEM-α containing 5% FCS, 1% Penicillin/Streptomycin (P/S, Thermo Scientific, Waltham, MA, USA), 10 mM calcium chloride (Carl Roth, Karlsruhe, Germany)) was mixed in a 1:1 ratio (120 µL total volume) with human peripheral blood. Blood was drawn max. At 2 h prior to clot formation and from healthy male volunteers using EDTA monovettes. Both smokers’ and non-smokers’ blood was drawn from healthy (no secondary disease, healthy BMI) male volunteers. Smokers were on average 33 years old, non-smokers 31. The mixture was allowed to coagulate for 1 h at 37 °C, humidified atmosphere, 5% CO_2_. Then, formed clots were transferred to fresh 96-well plates, and 100 µL culture medium (MEM-α, 5% FCS, 1% P/S) were added. In vitro hematomas were cultured within a hypoxia incubator chamber (STEMCELL Technologies, Vancouver, BC, Canada) filled with a hypoxic gas mixture (1% O_2_, 5% CO_2_, 94% N_2_ (Westfalen, Münster, Germany)) in a humidified atmosphere for up to 48 h.

### 2.3. Simulation of Smoking Conditions In Vitro

For simulation of smoking conditions in vitro, a combination of smokers’ blood and Cigarette Smoke Extract (CSE) pre-stimulated SCP-1 cells was used. For stimulation of SCP-1 cells, CSE was prepared as described [25]. A 5% CSE (corresponds to 5 cigarettes per day) value was used for pre-stimulation as this previously has been shown to significantly impact the cells’ function [26]. SCP-1 cells were pre-stimulated for 7 days. The medium was changed every 2 to 3 days. Stimulation was renewed with each medium change. Experimental setup is further visualized in Figure 1.

### 2.4. Life Staining

Hematomas were stained with a 0.1% CalceinAM (Biomol, Hamburg, Germany) solution in culture medium for 20 min, 37 °C, dark. The stain was removed, and in vitro hematomas were washed with PBS once. Images were taken using an EVOS FL Cell imaging system. Clot diameters were determined using ImageJ (Bethesda, MD, USA).

### 2.5. Histology

For histological analyses, in vitro hematomas were fixed in 4% formaldehyde, embedded in paraffin, cut in 4 µm-thick slices, and mounted on glass slides (Superfrost Plus, Fisher Scientific, Schwerte, Germany). Sections were stained with hematoxylin–eosin and Movat pentachrome stains [27] and evaluated using a Leica DM6000B light microscope (Leica, Wetzlar, Germany). Image acquisition was performed using the Diskus Mikroskopische Diskussion image acquisition software (Carl H. Hilgers, Koenigswinter, Germany).

### 2.6. Sex-Specific Polymerase Chain Reaction

To determine the ratio of incorporated SCP-1 to blood cells, sex-specific qPCR was performed as previously described [28]. Since the SCP-1 cells were derived from a female, all blood donors were male. DNA was isolated via alkaline lysis using a 50 mM sodium hydroxide solution (98 °C, 20 min) followed by neutralization with an equal amount of 100 mM Tris, pH = 8. After isolation, DNA was frozen immediately at −80 °C. The total amount of DNA was determined by detecting UDP Glucuronosyltransferase Family 1 Member A6 (*UGT1A6*) gene, whereas male DNA content was assessed by detecting y-chromosomal specific sex-determining region Y (*SRY*) gene. Sample-specific DNA content was determined by respective standard curves. PCRs were run in total volumes of 20 µL containing 10 µL Green Master Mix (2X) High Rox (Biozym, Hessisch Oldendorf, Germany), 1 µL primer forward, 1 µL primer reverse, 7 µL of RNase/DNase-free water, and 1 µL of template DNA. The cycling program included 15 min initial denaturation at 95 °C followed by 40 cycles of denaturation (95 °C, 30 s), primer annealing (60 °C, 30 s), and elongation (72 °C, 30 s), and a final elongation (72 °C, 15 min). Melting curves of amplified DNA were recorded to ensure specific amplification only. PCRs were run within a StepOnePlus™ Real-Time PCR System (Thermo Scientific, Waltham, MA, USA) and analyzed using the StepOne Software v2.3 (Thermo Scientific, Waltham, MA, USA), where cycle threshold (C_T_) and baseline were set automatically. Primer details are listed in Table 1. Experiments were performed for each donor in three technical replicates (N = 5, n = 3) and standard curves in three biological and technical replicates (N = 3, n = 3). Standard curves and additional experimental details are shown in Appendix A.

### 2.7. Mitochondrial Activity (Resazurin Conversion)

Mitochondrial activity was assessed through Resazurin conversion. In brief, 100 µL of 0.001% Resazurin working solution (Sigma-Aldrich/Merck, Darmstadt, Germany) was added per well or in vitro hematoma. After incubation at 37 °C, formed Resorufin was measured at λEx = 544 nm/λEm = 590–10 nm using the Omega plate reader.

### 2.8. Adenosine Triphosphate Content

Adenosine triphosphate (ATP) content was determined by using the CellTiter-Glo^®^ 2.0 Cell Viability Assay (Promega, Madison, WI, USA) following the manufacturer’s instructions. For analysis of 3D cultures, the incubation time was increased to 30 min. Luminescence was quantified using the Omega plate reader.

### 2.9. Lactate Dehydrogenase Release

Release of lactate dehydrogenase (LDH) in the culture supernatant was determined by CyQUANT™ LDH Cytotoxicity Assay (Thermo Scientific, Waltham, MA, USA) following the manufacturer’s instructions using 50 µL culture supernatant per technical replicate.

### 2.10. Enzyme-Linked-Immunosorbent-Assay

Levels of IL-6, TNF-α, and CCL2 in the culture supernatant were determined by enzyme-linked-immunosorbent-assay (ELISA) following the manufacturer’s instructions (Peprotech, Hamburg, Germany).

### 2.11. Alkaline Phosphatase Activity

As a measure for osteogenic differentiation, alkaline phosphatase (ALP) activity, was determined by conversion of 4-Nitrophenyl phosphate (pNpp) to 4-Nitrophenol (pNp) [29]. A 200 µL measure of pNpp substrate solution (3.5 mM pNpp in 50 mM Glycine, 100 mM TRIS, 1 mM MgCl_2_, pH 10.5 (all Carl Roth, Karlsruhe, Germany)) was added per in vitro hematoma and incubated at 37 °C for 1 h. Clots were removed from the solution and absorbance at λ = 405 nm was measured using the Omega Plate reader. ALP activity was normalized to mitochondrial activity.

### 2.12. Gene Expression Analysis

RNA was isolated from a pool of 12 in vitro hematomas per condition using Chloroform/Phenol extraction. Prior to RNA isolation, in vitro hematomas were homogenized in RNA isolation solution using homogenization pestles. cDNA synthesis was performed using the RevertAid First Strand cDNA Synthesis Kit (Thermo Scientific, Waltham, MA, USA) following the manufacturer’s instructions. Reverse transcription-polymerase chain reactions (RT-PCRs) were conducted using a total reaction volume of 15 µL consisting of 7.5 µL Red HS Master Mix (Biozym, Hessisch Oldendorf, Germany), 0.75 µL primer forward and reverse, respectively, 2 µL template cDNA, and 4 µL RNase/DNase free water. PCRs were run within an Arktik™ Thermocycler (Thermo Scientific, Waltham, MA, USA) with 2 min, 95 °C initial denaturation; between 23 and 40 cycles of 15 s of each denaturation (95 °C), primer annealing (52–64 °C), elongation (72 °C); and a final elongation step (10 min, 72 °C). Detailed information on primers used and the cycling conditions are listed in Table 2. For visualization, PCRs were applied on 1.8% Agarose (Genaxxon, Ulm, Germany) gels containing 0.007% ethidium bromide (Carl Roth, Karlsruhe, Germany). Images were captured with IntasGelDoc (INTAS, Göttingen, Germany) and analyzed using the ImageJ Gel Analysis Tool. Gene expression results were normalized to housekeeping gene EF1α and to the expression of the control C1 at 4 h.

### 2.13. Angiogenesis Array

Human angiogenesis array C1000 (RayBiotech, Peachtree Corners, GA, USA) was performed following the manufacturer’s instructions, using pooled culture supernatants collected after 48 h of culture. Chemiluminescent signals were recorded using an INTAS Chemocam (Göttingen, Germany) and analyzed using ImageJ. Obtained data were normalized to the internal controls.

### 2.14. Culture of Human Umbilical Vein Endothelial Cells

Human umbilical vein endothelial cells (HUVECs) were cultured in Endothelial Cell Growth Basal Medium 2 (EBM-2, Peprotech, Hamburg, Germany) supplemented with 2% FCS, 1% Antibiotic/Antimycotic (A/A, PAA Laboratories, Toronto, ON, Canada), 0.5 ng/mL human VEGFA165, 10 ng/mL human FGF-b, 5 ng/mL human Epidermal growth factor (EGF), 20 ng/mL hRIGF-R3 (all Peprotech, Hamburg, Germany), 22.5 µg/mL Heparin (Leo Pharma, Bellerup, Denmark), 0.2 µg/mL Hydrocortisone (Pfizer, New York, NY, USA), and 1 µg/mL L-ascorbic acid (Sigma-Aldrich/Merck, Darmstadt, Germany). HUVECs were cultured on 0.1% Gelatin (Sigma-Aldrich/Merck, Darmstadt, Germany)-coated flasks and subcultured when confluent. The medium was changed every 2 to 3 days. For experiments, cells were used in passages 7 to 10.

### 2.15. HUVEC Proliferation Assay

HUVECs were seeded on with 0.1% Gelatin-pre-coated 96-well plates. Cells were stimulated with a 1:1 mixture of culture supernatant and EBM-2. Proliferation was indirectly assessed after 48 h by means of mitochondrial activity and ATP content as described in Section 2.7 and Section 2.8. Data were normalized to Ctrl (C1) for simplification.

### 2.16. Tube Formation Assay

48-well plates were coated with 6 µL GeltrexTM (Thermo Scientific, Waltham, MA, USA) using a thin-layer angiogenesis assay method [30]. A total of 6.5 × 10^4^ HUVECs were seeded per well and the respective stimuli were added 1:1 diluted in plain EBM-2 to a total volume of 200 µL. Tube formation assays were performed using pooled culture supernatants. Cells were incubated in humidified atmosphere, 37 °C, 5% CO_2_ for 18 h. Before microscopy, tubes were stained with 0.1% CalceinAM for 10 min, 37 °C. Microscopic images were captured using an EVOS FL imaging system (Thermo Scientific, Waltham, MA, USA) and analyzed using the ImageJ angiogenesis analyzer [31]. Data were normalized to Ctrl (C1) for simplification.

### 2.17. Statistics

Statistics were made using Graph Pad Prism 8 (San Diego, CA, USA). When comparing solely two experimental groups, data were compared using non-parametric Mann–Whitney tests. If more than two experimental groups were compared, non-parametric Kruskal–Wallis tests followed by Dunns’ multiple comparison tests were performed. The number of biological replicates (N) and technical replicates (n) is given in the figure legend for each experiment performed. Levels of significances were defined as * *p* < 0.05, ** *p* < 0.01, *** *p* < 0.001. Data are shown as box plots showing all data points if not indicated differently.

## 3. Results

As described in the materials and methods section, experiments were performed using blood from non-smokers and smokers paired with SCP-1 cells either pre-stimulated with CSE or not. As active smokers not only show a diverging blood composition but also impaired mesenchymal stem cells, two main experimental conditions were defined: smokers’ blood and CSE-pre-stimulated SCP-1 cells mimicking the smokers’ hematomas (S) and non-smokers’ blood combined with unstimulated SCP-1 cells mimicking the non-smokers hematomas (Ctrl). The respective control conditions, combining smokers’ blood with unstimulated SCP-1 cells (E1) and non-smokers’ blood with pre-stimulated SCP-1 cells (C2), were also measured. As the main part of the manuscript solely focuses on the two experimental groups, results of the control conditions are displayed in Appendix A.

### 3.1. Smokers’ and Non-Smokers’ Hematomas Show Comparable Cell Viability and Survival

First, smokers’ and non-smokers’ hematomas were compared in terms of viability and stability to allow later comparison of results. As shown in Figure 2, smoker hematomas showed a slight but not significant increase in mitochondrial activity, whereas no differences in ATP content and ratio between blood and SCP-1 cells were observed. As expected, smoker hematomas exhibited a higher release of LDH. In addition, higher clot diameters were detected in smokers’ hematomas. Hematoxylin–eosin and Movat pentachrome staining of formaldehyde-fixed, paraffin-embedded hematomas showed a comparable number and distribution of SCP-1 cells in a background of erythrocytes. Additionally, fibrin as well as spindle cell differentiation of SCP cells in both hematomas from non-smokers as well as from smokers could be detected (see Figure 2g). In conclusion, smokers’ hematomas were as stable as nonsmokers’ hematomas and thus can be used comparably in further experiments.

### 3.2. Smokers’ Hematomas Show Increased Inflammation and Decreased Osteogenic and Chondrogenic Differentiation Potential

The inflammatory status of hematomas was determined by expression and release of CCL2, TNF-α, and IL-6. The results are shown in Figure 3. Expression and release of CCL2 was significantly increased in smoking conditions when compared to control conditions. Although the amount of secreted TNF-α was surprisingly low, we could detect an increase in expression of TNF-α after 4 h which persisted until 48 h. IL-6 secretion, as well as expression, showed a trend towards higher occurrence in smokers’ hematomas; nevertheless, results were not significant. Summarizing the results, we could show that smokers’ hematomas show a higher inflammatory status. When comparing the hematoma control conditions with the respective other mesenchymal stem cell type, one could further see that the inflammatory status was strongly dependent on the blood donor and not the pre-stimulation of the SCP-1 cells (see Appendix A).

As a marker of early osteogenic differentiation, ALP activity was determined after 48 h, as can be seen in Figure 4. ALP activity was significantly downregulated in smokers’ hematomas. In line with this, the expression of osteogenic differentiation markers *RUNX2, ALP, BMP4*, and *Noggin* as well as the chondrogenic differentiation marker *SOX9* were all significantly lower in smokers’ hematomas (comp. Figure 4b). After comparing the control conditions with the respective other cell type shown in Appendix A, downregulation of factors in the smokers’ hematomas seems to be a two-sided effect driven by the smokers’ blood and the pre-stimulation of the SCP-1 cells. Solely *BMP2* expression showed no significant difference in expression between the two groups. In summary, we can conclude that the hematomas of smokers show a drastic reduction in osteogenic and chondrogenic potential.

### 3.3. Smoking Negatively Affects Angiogenic Stimuli in In Vitro Fracture Repair

The hematomas’ potential to induce angiogenesis under smoking and non-smoking conditions was analyzed using an angiogenesis array. Results are displayed in Figure 5 Smokers’ hematomas showed a higher release of inflammatory factors; however, the effect was rather small. The only exception was TNF-α, which was not detectable in culture supernatants from non-smokers’ hematomas but strongly secreted by smokers’ hematomas. Of the growth factors EGF, Granulocyte-macrophage colony-stimulating factor (GM-CSF), and PDGF-BB, only GM-CSF showed increased release in non-smokers’ hematomas when compared to smokers’ hematomas. Smokers’ hematomas further secreted higher levels of Tissue inhibitor of metalloproteases (TIMP) 1/2 and lower levels of MMP1 and 9. Surprisingly secretion of VEGF-A, VEGF-D, and Leptin was increased in the smokers’ hematomas, but VEGFR2/3, Angiopoietin 1/2 (Angpt1/2), and its receptor Tie2 were decreased drastically. Further secretion of angiogenesis inhibitor endostatin was lower in smokers’ than in non-smokers’ hematomas. Results of main influencing factors *PDGF-BB*, *VEGFA*, *Angpt1*, *MMP9*, and *GM-CSF* were additionally confirmed by gene expression analysis, whereas except for *GM-CSF* all genes were expressed less in the smokers’ hematomas.

As secretion and expression of pro-angiogenic factors of smokers’ hematomas was reduced, the angiogenic ability of hematomas was assessed via stimulation of HUVEC cells with hematomas culture supernatants in a proliferation and tube formation assay. As can be seen in Figure 6 smokers’ hematomas supernatants were less capable of inducing HUVEC proliferation, as determined by mitochondrial activity and ATP content. Further HUVEC tube formation with smokers’ hematomas’ supernatants was diminished. HUVECs build fewer and shorter tubes and smaller networks (Figure 5b). Fewer junctions and a smaller mesh area and, in line with that, higher numbers of isolated segments, were detected when stimulating with smokers’ hematomas’ supernatants. In conclusion, smokers’ hematomas produced fewer pro-angiogenic factors and were less capable of inducing angiogenesis in vitro.

## 4. Discussion

The bone is an overly complex and dynamic organ. It is constantly reshaped by the balanced activity and presence of the cells found in the tissue. In the case of injury, precise coordination of reactions is required, which can be easily disturbed when the bone metabolism is affected by other conditions or habits such as cigarette smoking [32,33]. To model early fracture repair, we aimed to adapt a recently published in vitro hematoma model for additionally displaying the vascular system as a mandatory contributor to appropriate fracture repair [22,34]. By analyzing non-smokers’ and smokers’ hematomas, we proved the clinical relevance of the model and identified diverging reactions in smokers.

Overall, the viability of smokers’ and non-smokers’ hematomas was comparable, and CSE pre-stimulation of SCP-1 cells did not affect proliferation (see Figure 1 and Appendix A). Slightly higher mitochondrial activities in smokers’ hematomas can be explained by oxidative damage due to previously shown CSE-induced ROS production in hMSCs [25] and smokers’ blood [35]. It must be noted that, due to technical limitations, detection of ROS in in vitro hematomas was not possible. Increased LDH levels are no reason for concern, as they are a common phenomenon due to higher tissue damage in smokers [36].

In vitro hematomas produced with smokers’ blood were larger in size than non-smokers hematomas, indicating greater clotting and a diverging fibrin structure. It is well known that smokers tend to have a more rapid clotting of blood, e.g., due to higher fibrinogen levels or faster rapid platelet aggregation, increasing the risk of smokers’ developing, for instance, atherosclerosis or thrombosis. In previous in vitro studies, the coagulation structure of smokers was described as reticular with some areas of plaque formation [37], and, upon acute CSE stimulation, formed clots were higher in clot strength [38]. Histologic analyses showed no difference in the overall amount of fibrin between smokers’ and non-smokers’ in vitro hematomas; however the possibility to visualize fibrin organization using routine histology is limited. Further investigations are needed to determine whether the fibrin structure is altered and how the clot architecture finally affects fracture repair.

In both of our models, smokers’ and non-smokers’ in vitro hematomas showed high levels of cytokine secretion and expression, nicely reflecting the inflammatory phase of healing [22]. One general contributor to inflammation in the here described system is the hypoxic environment, which we and others could previously show drastically increased inflammation in comparable model systems [23,39]. Smokers’ hematomas exhibited a higher and prolonged inflammatory state as compared to non-smokers’ hematomas (see Figure 1 and Figure 4a). This is not surprising, as in vivo smokers suffer from constant tissue hypoxia due to higher carbon monoxide levels reducing overall oxygen binding to erythrocytes [40]. The resulting oxidative stress is contributing to inflammation, represented by increased synthesis of inflammatory cytokines such as CCL2 [41]. Within the course of our experiments, we could detect that the higher inflammatory state mainly resulted from the blood used and not from the pre-stimulation of the SCP-1 cells (see Appendix A). This is consistent with previous studies showing a chronically inflamed state of the blood of smokers by higher cytokine levels e.g., IL-6, IL-8, and carcinoembryonic antigen (CEA) [42], and a higher number of immune cells, especially monocytes, granulocytes, and leukocytes [42,43].

Within our in vitro hematomas, smoking affected the osteogenic ability as seen by a decrease in ALP activity, and lowered expression of all osteogenic/chondrogenic differentiation markers after 48 h of incubation. Additionally, in our model, both CSE pre-stimulation and smokers’ blood contributed to diminished differentiation (see Appendix A). Results are in line with our previous studies showing that osteogenesis in CSE-stimulated MSCs is diminished through ROS formation [25]. Remarkably, a reduction in osteogenic potential was observed despite the presence of antioxidants such as albumin and glutathione in blood, which can partially neutralize formed ROS [44]. When analyzing fracture callus and non-union tissue, Kloen et al. showed a decreased availability of BMPs and their inhibitors including Noggin within the non-union group, pointing out that smokers’ hematomas showed early signs of delayed healing [45]. Furthermore, smokers’ higher inflammatory status can negatively influence osteogenesis. Lu et al. reported that bone remodeling in mice was diminished by inflammation and following NF-kB activation by decreasing osteoblastic differentiation and increasing osteoclast formation [46].

In respect to angiogenesis, smokers’ hematomas secreted lower levels of MMPs and higher levels of TIMPs, whereas VEGFA and D release were enlarged. Contrarily, secretion of VEGFR2/3 receptor, Angpt1 and 2, and their receptor Tie2 was reduced in smokers’ hematomas. The most studied and important endothelial stimulus is undoubtedly VEGF. VEGF family members exert their action via binding to VEGF-Receptors (VEGFRs) on the cell surface. VEGFRs are receptor tyrosine kinases and ligand binding triggers the phosphorylation of intracellular tyrosine residues which can activate various signaling pathways including phospholipase Cγ (PLCγ)–ERK1/2 pathway, the Pi3K-AKT-mTOR pathway, and SRC and small GTPases [47]. Mainly expressed in endothelial cells, VEGFRs are also expressed in monocytes, macrophages, hematopoietic stem cells, and MSCs, which explains their presence in our fracture hematomas [48,49]. Whereas VEGFR1/2 expression in MSCs increased under hypoxic conditions and can be related to osteoblastic differentiation [50], within the growth plate of bones VEGFD was found to bind to VEGFR3 in osteoblasts, suggesting a role in intramembranous bone formation [51]. Due to the lack of endothelial cells, decreased detection of VEGFRs within our model may rather indicate a decrease in osteogenic differentiation as in angiogenesis. In endothelial cells, VEGFR signaling contributes to endothelial cell survival, proliferation, and migration, and can increase vascular permeability. It is therefore associated with vessel sprouting and initial angiogenic events [52,53]. In bone, VEGF also directly stimulates endothelial cells to generate osteogenic stimuli through an angiocrine effect and is therefore crucial for endothelial–bone communication [54]. During fracture repair, VEGF is an essential growth factor. Its inhibition in mice delayed fracture repair and resulted in insufficient vasculature [55,56]. Mice which were exposed to cigarette smoke showed lower protein levels of VEGF and vWF in fracture calluses combined with higher rates of fracture non-unions [56]. In our smokers’ hematomas, we detected a slight reduction in VEGFA expression correlating to the previously highlighted study. Nevertheless, especially the secretion of VEGFD was higher in smokers’ blood hematomas, whereas VEGFA secretion could be additionally promoted by CSE pre-stimulation (see Appendix A). In literature, serum levels of VEGF in smokers have been described as both increased and decreased. Köttstrofer et al. reported reduced VEGFA and M-CSF concentrations in smokers [57], whereas Ugur et al. showed the opposite [58]. Although VEGF is so crucial in angiogenesis, massive VEGF production may also be associated with increased bone resorption and immature vessel formation [59]. Further sufficient vasculature does not only rely on VEGF alone. Wang et al. show that in an arthritis model under inflammatory conditions higher VEGFA levels were detected. Despite these elevated VEGFA levels, the mice were suffering from delayed healing, reflecting our results [60].

In contrast to VEGF, which induces vessel sprouting and early angiogenesis, angiopoietins, especially Angpt1, are described to be important modulators in angiogenic maturation and development [61]. Both Angpt1 and 2 bind to the receptor Tie-2, whereas Angpt1 acts as an agonist and Angpt2 as an antagonist [62]. Angpt1 leads to maturation and stabilization of the microvasculature [63] and has been described to positively influence osteogenesis [64]. In contrast, Angpt2 in the presence of VEGFA leads to vascular sprouting [65]. Angpt1 levels are described as fairly constant, whereas Angpt2 expression fluctuates and can be induced by hypoxia [66] or inflammatory mediators, e.g., thrombin [67], which makes its expression likely within the fracture environment. Within growing bone, Angpt1 and 2 are co-expressed with VEGFA mainly in osteoblasts [68]. Within a sheep delayed fracture healing model, both Angpt1 and 2 are downregulated [69]. In bone defects in rabbits, endogenous Angpt2 promoted vascularization and osteogenesis via autophagy, leading to a faster regeneration [70]. Lower levels of Angiopoietins and Tie2 in smokers are therefore likely to be an important part of the observed reduced angiogenic potential of smokers’ hematomas.

Additionally, also the reduction in MMPs and the increase in TIMPs in smokers’ early fracture repair point out the reduced ability for matrix remodeling and therefore angiogenesis. MMPs are involved in leveling the path for the ingrowth of new blood vessels but also help in callus remodeling and increase the bioavailability of different growth factors stored in the ECM [71]. This also represents the major limitation of the here-presented model which allows investigation of the influence of the fracture hematoma on the endothelium in a single-sided mode and not *vice versa*. However, osteogenesis, angiogenesis, and inflammation are highly interdependent and it is known that also the endothelium stimulates osteogenesis, for instance via endothelial Notch, which has been shown to promote both angiogenesis and osteogenesis [72]. However, an improved version of the model is required to consider such a direct interaction. Higher TIMP1 levels were also detected in a sheep delayed fracture healing model [69]. In contrast, in non-union patients, lower TIMP1 and TIMP2 levels, but similar MMP9 levels, were detected as in normal fracture repair, but results were obtained 1 to 24 weeks after fracture and therefore not ultimately comparable with our model [73]. The exact mechanism behind the reduced angiogenesis still needs to be established in more detail. In summary, our model showed that increased inflammation and reduced osteogenic potential seem to dysregulate early angiogenesis in smokers by increased VEGF levels but decreased Angpt1/2 and its receptor Tie-2 as well as reduced potential for in matrix remodeling. Therefore, our smokers’ hematomas show already in this early stage of fracture repair obvious signs for a potential delay in healing. Thus, our model system could function as a drug-testing system to facilitate fracture healing in smokers.

## Figures and Tables

**Figure 1 bioengineering-09-00186-f001:**
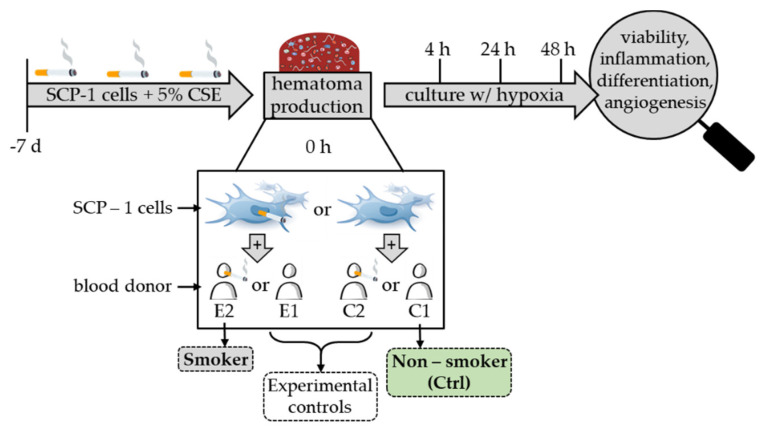
Experimental setup and conditions.

**Figure 2 bioengineering-09-00186-f002:**
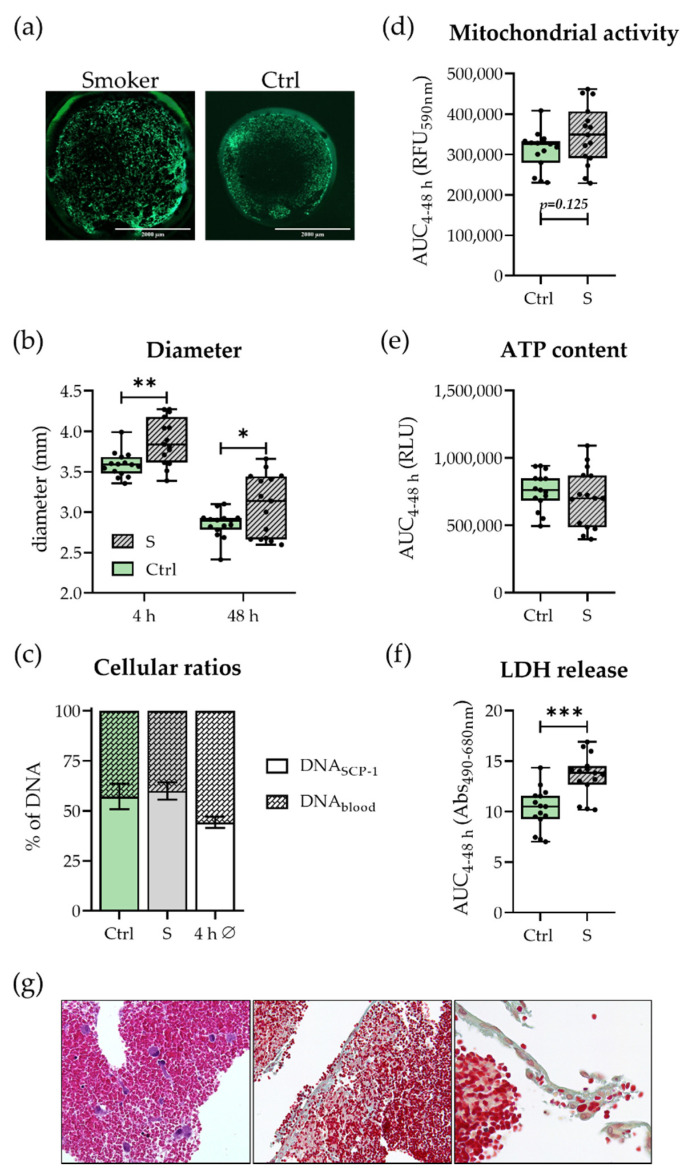
Viability of smokers’ (S) and non-smokers’ (Ctrl) in vitro hematomas. With (**a**) Exemplary life staining images after 48 h in 1.25× magnification. Scale bars refer to 2000 µm. (**b**) Hematoma diameter after 4 and 48 h of incubation. (**c**) Ratio between SCP-1 cells and blood cells determined by sex-specific qPCR after 48 h compared to the average after 4 h (4 h Ø). Data are shown as mean ± SEM. (**d**) Mitochondrial activity determined by Resazurin conversion. (**e**) ATP content in the hematomas. (**f**) LDH release into the culture supernatant. (**d**–**f**) are all represented as Area Under the Curve (AUC) over the entire cultivation period of 48 h. (**g**) Representative microphotographs from formaldehyde-fixed, paraffin-embedded in vitro hematomas: left image (non-smoker, hematoxylin-eosin), highlighted SCP-1 cells (arrow); center/right image (smoker, Movat pentachrome), erythrocytes, and fibrin, higher magnification in right image shows spindle cell differentiation of SCP-1 cells. Images were recorded in 200× (left and center) and 400× (right image) magnification. Experiments were performed in N = 5, n = 3. * *p* < 0.05, ** *p* < 0.01, *** *p* < 0.001.

**Figure 3 bioengineering-09-00186-f003:**
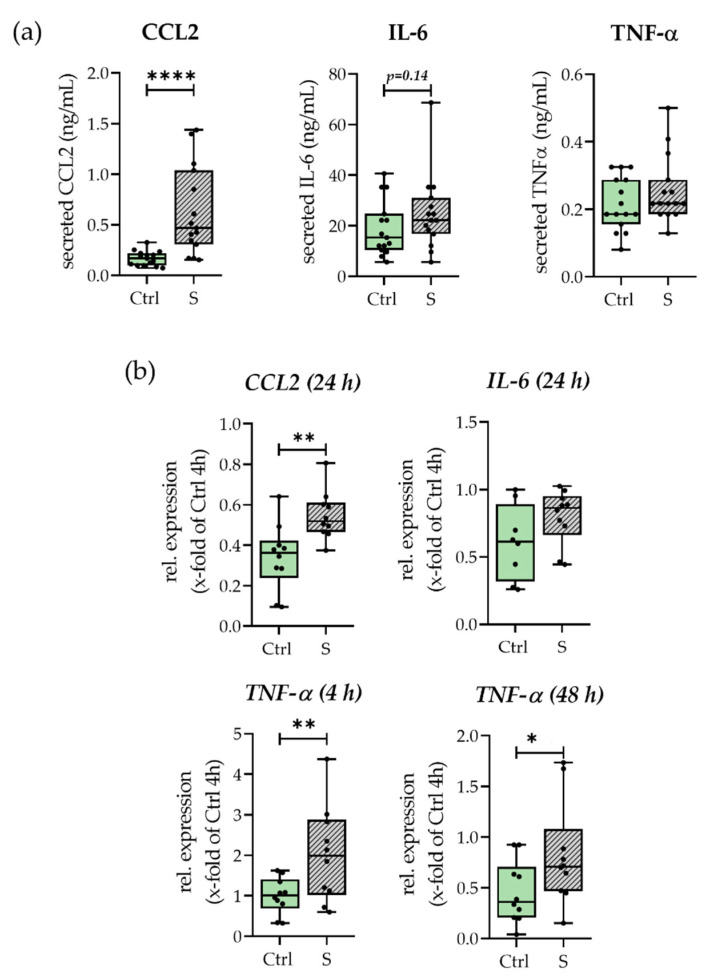
Influence of smoking conditions on inflammation. With (**a**) Secretion of CCL2, TNF-α, and IL-6 after 48 h. N = 5, n = 3. (**b**) Expression of *IL-6*, *TNF-α*, and *CCL2*. Shown time points are indicated in graphs. N = 5, n = 2. * *p* < 0.05, ** *p* < 0.01, **** *p* < 0.0001.

**Figure 4 bioengineering-09-00186-f004:**
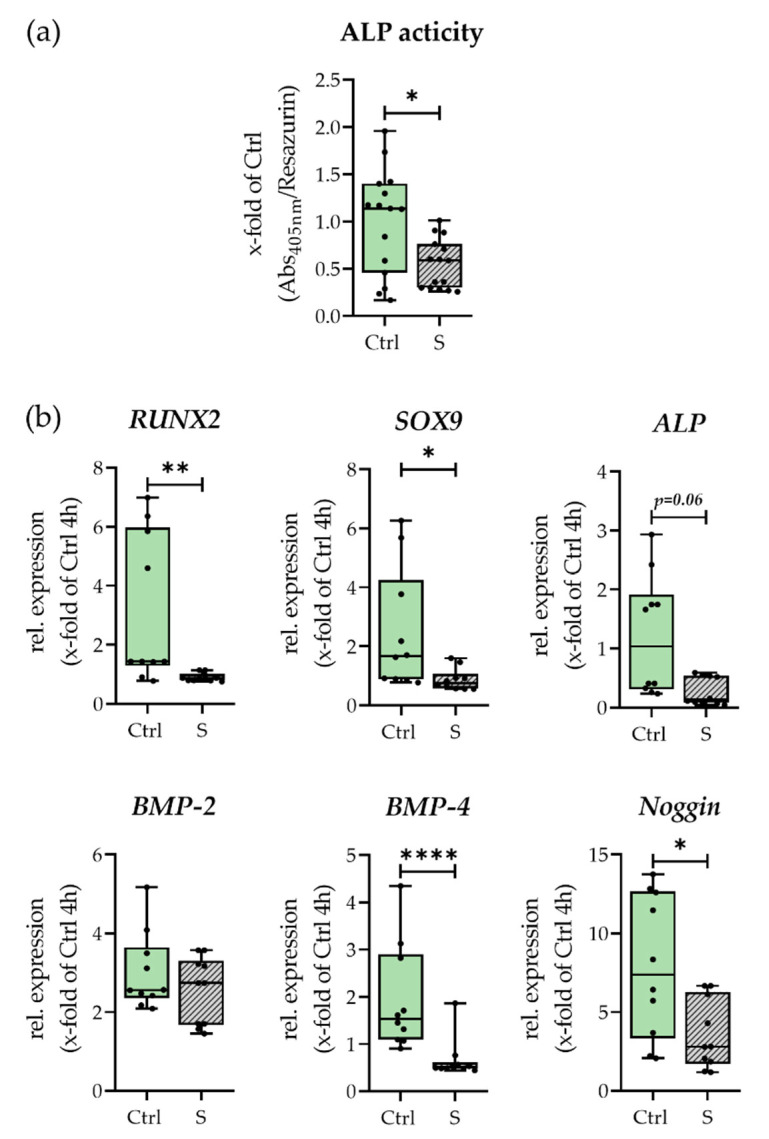
Osteogenic and chondrogenic differentiation potential of smokers’ (S) and non-smokers’ (Ctrl) in vitro fracture hematomas. With (**a**) ALP activity after 48 h. N = 5, n = 3. (**b**) Expression of *RUNX2, SOX9, ALP, BMP2, BMP4,* and *Noggin* after 48 h. N = 5, n = 2. * *p* < 0.05, ** *p* < 0.01, **** *p* < 0.0001.

**Figure 5 bioengineering-09-00186-f005:**
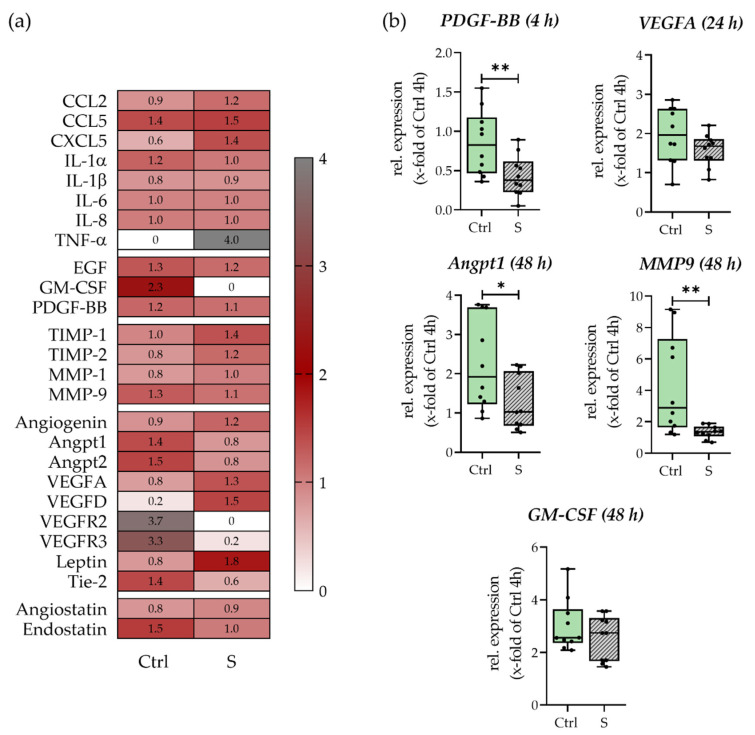
Analysis of angiogenesis-stimulating factors of smokers’ (S) and non-smokers’ (Ctrl) in vitro fracture hematomas. With (**a**) Secretion of angiogenic factors determined by angiogenesis array. Data are displayed normalized to the overall mean of each target. N = 5 (pooled), n = 4. (**b**) Gene expression of pro-angiogenic factors *VEGFA*, *Angpt1*, *MMP9*, *PDGF-BB*, and *GM-CSF*. N = 5, n = 2. * *p* < 0.05, ** *p* < 0.01.

**Figure 6 bioengineering-09-00186-f006:**
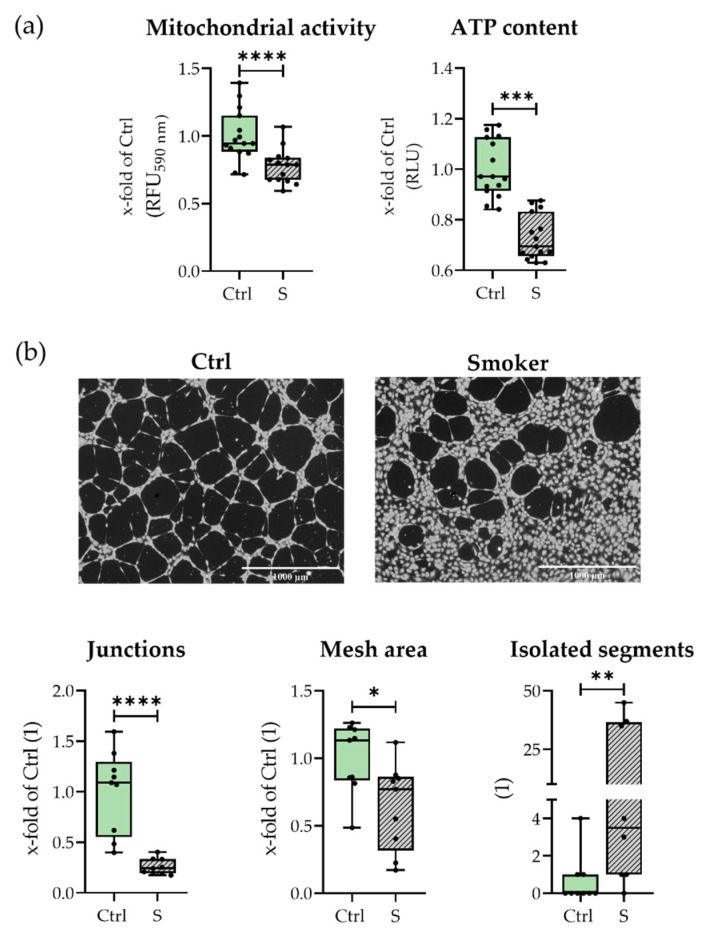
Effect of smokers’ (S) and non-smokers’ (Ctrl) culture supernatant on HUVEC cells. With (**a**) Effect on HUVEC proliferation measured by mitochondrial activity and ATP content in N = 5, n = 3. Both showed as x-fold of Ctrl; (**b**) Effect on HUVEC tube formation with representative life staining of formed tubes and number of junctions (as x-fold of Ctrl), Mesh area (as x-fold of Ctrl), and number of isolated segments. Shown images were recorded in 4× magnification. Scales bars indicate 1000 µm length. N = 3, n = 3. * *p* < 0.05, ** *p* < 0.01, *** *p* < 0.001, **** *p* < 0.0001.

**Table 1 bioengineering-09-00186-t001:** Primer details sex-specific polymerase chain reaction.

Target	Gene Symbol	Primer Forward	Primer Reverse	Fragment Size	T_An_ ^1^
*UGT1A6*	NC_000002.12	TGGTGCCTGAAGTTAATTTGCT	GCTCTGGCAGTTGATGAAGTA	209 bp	60 °C
*SRY*	NC_000024.10	TGGCGATTAAGTCAAATTCGC	CCCCCTAGTACCCTGACAATGTATT	137 bp	60 °C

^1^ T_An_: Annealing Temperature.

**Table 2 bioengineering-09-00186-t002:** Primer details and cycling conditions.

Target	Primer Forward	Primer Reverse	Amplicon Size	T_An_ ^1^	n_cycles_ ^2^
*ALP*	ACGTGGCTAAGAATGTCATC	CTGGTAGGCGATGTCCTTA	476 bp	53 °C	40
*Angpt1*	CGATGGCAACTGTCGTGAGA	CATGGTAGCCGTGTGGTTCT	232 bp	60 °C	35
*BMP2*	CCCCCTACATGCTAGACCTGT	CACTCGTTTCTGGTAGTTCTTCC	150 bp	60 °C	35
*BMP4*	TGGTCTTGAGTATCCTGAGCG	GCTGAGGTTAAAGAGGAAACGA	130 bp	60 °C	40
*CCL2*	CCTTCATTCCCCAAGGGCTC	GGTTTGCTTGTCCAGGTGGT	236 bp	60 °C	27
*EF1α*	CCCCGACACAGTAGCATTTG	TGACTTTCCATCCCTTGAACC	98 bp	56 °C	25
*FGF-2*	GGAGAAGAGCGACCCTCACA	TCATCCGTAACACATTTAGAAGCC	141 bp	60 °C	30
*GM-CSF*	GAGACACTGCTGCTGAGATGA	GAGGGCAGTGCTGCTTGTA	180 bp	64 °C	35
*IL-6*	AACCTGAACCTTCCAAAGATGG	TCTGGCTTGTTCCTCACTACT	159 bp	58 °C	30
*MMP9*	TCTATGGTCCTCGCCCTGAA	CATCGTCCACCGGACTCAAA	219 bp	60 °C	35
*Noggin*	CAGCGACAACCTGCCCCTGG	GATCTCGCTCGGCATGGCCC	250 bp	59 °C	33
*PDGF-BB*	CCAGGTGAGAAAGATCGAGATTG	ATGCGTGTGCTTGAATTTCCG	238 bp	60 °C	35
*RUNX2*	CTGTGGTTACTGTCATGGCG	GGGAGGATTTGTGAAGACGGT	170 bp	60 °C	30
*SOX9*	GAAGGACCACCCGGATTACA	GCCTTGAAGATGGCGTTGG	120 bp	60 °C	35
*TNF-α*	ATGAGCACTGAAAGCATGATCC	GAGGGCTGATTAGAGAGAGGTC	217 bp	59 °C	35
*VEGFA*	CTACCTCCACCATGCCAAGT	GCAGTAGCTGCGCTGATAGA	109 bp	60 °C	30

^1^ T_An_: Annealing Temperature. ^2^ n_cycles_: Number of cycles.

## Data Availability

The datasets generated during and/or analyzed during the current study are available from the corresponding author on reasonable request.

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
