# Peer review of "Smoking Impairs Hematoma Formation and Dysregulates Angiogenesis as the First Steps of Fracture Healing"

_bioengineering, 2022, doi:10.3390/bioengineering9050186_

Round 1
Reviewer 1 Report
The authors investigated in an in vitro hematoma model the influence of smoking on different outcomes (inflammation, osteogenic differentiation, angiogenic effects). The research question, as well as the used methods are interesting and appropriate. The manuscript is well written. However, I have some major points regarding data presentation and analysis, which needs to be addressed before publication.
- The authors sometimes present the data as x-fold to control and sometimes report the true outcome measures (like pg/ml). I would strongly recommend to always present the "real" data without further normalization. The only exception is of course qPCR data, were normalization is needed. But also here, the controls needs to be justified further or described in the methods (4h control as the reference, this is not described in the methods?). The same should be applied also for the data in the supplement. There is also some normalization going on I don't fully understand.
- I think the data in the supplement are really important to judge if the influence of the blood or the CSE is higher. To really be able to understand that, the authors should present all 4 groups (control, smoking and the two mixed groups) together in the supplement. I understand that because of statistical or clarity issues only two groups were chosen for the main figures, but this should be added to the supplement and discussed a bit further in the manuscript.
- In general, I would recommend to also show the single data points in the figures because the sample size is not clear for the groups. F.ex. Fig. 2:
"Experiments were performed in N=5, n=3."
What does that mean regarding the number of data points included in the figures? - Minor point: please describe how many blood donors were used and describe their main characteristics beyond smoking (age, BMI, health, etc.)
Reviewer 2 Report
The manuscript entitled "Smoking impairs hematoma formation and dysregulates angiogenesis as the first steps of fracture healing" is an interesting study performed with an original approach.
In this paper, authors describe a model aimed at showing in vitro the negative impact of smoking on early vascularization of newly formed bone tissue.
The paper has some strengths, but there are some critical points especially concerning gene expression experiments.
Introduction
Lines 35-37. At the beginning of the Introduction section, authors focus their attention on the delayed fracture healing that is one of the main causes of prolonged hospital stays. Consequently, prolonged hospitalization involves burden for the patient and high costs for the health care system.
After this part, authors should mention strategies/products conventionally used by orthopaedists to promote bone healing and consequently reduce the length of stay in hospital.
Concerning products, many companies have in their portfolio several medical devices claimed as alternatives to autologous bone transplantation. Moreover, some commercial products contain some growth factors useful for fracture healing successively described in the text (e.g., VEGF, BMP-2, PDGF-BB).
The following paper may help authors to have an overview about commercial products commonly used by clinicians to this aim:
Govoni M, Vivarelli L, Mazzotta A, Stagni C, Maso A, Dallari D. Commercial Bone Grafts Claimed as an Alternative to Autografts: Current Trends for Clinical Applications in Orthopaedics. Materials (Basel). 2021 Jun 14;14(12):3290. doi: 10.3390/ma14123290.
Lines 84-92. Please, replace the simple future tense with the simple past tense to use the same verb forms in this part of the text.
Materials and Methods
Line 96. Concerning SCP-1 cells, although authors have included a reference, more details about this single-cell-picked clone should be provided.
Moreover, authors should explain the rationale behind the use of this transduced cell population.
Lastly, please specify the passage number of SCP-1 cell cultures used for the preparation of in vitro hematomas and the simulation of smoking conditions.
Line 116. Authors assert that Cigarette Smoke Extract (CSE) was prepared as described in the Ref. 24. However, in this reference, 5% CSE corresponds to exposures associated with smoking up to 10 cigarettes/day and not to 5 per day, as reported in the text. Please, correct or better explain what is reported in this section.
Line 154. Please add information regarding Resazurin (Company, city, country).
2.6. Sex-specific Polymerase chain reaction. Authors report using qPCR to determine the ratio of incorporated SCP-1 to blood cells. Therefore, this reviewer expects that authors have performed a quantitative Real Time PCR, as suggested by the MIQE guidelines (The MIQE guidelines: minimum information for publication of quantitative real-time PCR experiments. Clin Chem. 2009 Apr;55(4):611-22. doi: 10.1373/clinchem.2008.112797).
As qPCR, please provide more methodological details.
2.12. Gene expression analysis. PCR has completely revolutionized the detection of RNA and DNA. Traditional PCR has advanced from detection at the end-point of the reaction to detection while the reaction is occurring. Moreover, traditional PCR has too many limitations: poor precision, low sensitivity, low resolution, are only some of critical issues related to the use of this old technique.
Therefore, in 2022, this reviewer would expect that traditional PCR, agarose gels containing ethidium bromide, and software used to capture and analyse images, are abandoned in favour of real time technologies. The same data of gene expression performed with real time PCR could achieve different results. However, taken together, considering all assays, results seem to be convincing, but this reviewer strongly recommends updating lab equipment.
Finally, a couple of suggestions:
- Please add “Germany” at the end of the affiliation number 1 reported after the manuscript title.
- In the Ref. 5, the date authors accessed the material is missing.
Round 2
Reviewer 1 Report
The authors addressed all of my previous concerns, thank you.
Reviewer 2 Report
The authors have addressed the issues raised previously, and the manuscript is suitable for publication in its current form.
Best regards